# Enhanced Mechanical Properties of Al_2_O_3_ Nanoceramics via Low Temperature Spark Plasma Sintering of Amorphous Powders

**DOI:** 10.3390/ma16165652

**Published:** 2023-08-17

**Authors:** Dongjiang Zhang, Rui Yu, Xuelei Feng, Xuncheng Guo, Yongkang Yang, Xiqing Xu

**Affiliations:** 1Xi’an Modern Control Technology Research Institute, Xi’an 710065, China; 2School of Materials Science & Engineering, Chang’an University, Xi’an 710061, China

**Keywords:** amorphous, Al_2_O_3_ nanoceramics, spark plasma sintering, low temperature sintering, transgranular fracture

## Abstract

In this work, Al_2_O_3_ nanoceramics were prepared by spark plasma sintering of amorphous powders and polycrystalline powders with similar particle sizes. Effective comparisons of sintering processes and ultimate products depending on starting powder conditions were explored. To ensure near-full density higher than 98% of the Al_2_O_3_ nanoceramics, the threshold temperature in SPS is 1450 °C for polycrystalline Al_2_O_3_ powders and 1300 °C for amorphous powders. The low SPS temperature for amorphous powders is attributed to the metastable state with high free energy of amorphous powders. The Al_2_O_3_ nanoceramics prepared by amorphous powders display a mean grain size of 170 nm, and superior mechanical properties, including high bending strength of 870 MPa, Vickers hardness of 20.5 GPa and fracture toughness of 4.3 MPa∙m^1/2^. Furthermore, the Al_2_O_3_ nanoceramics prepared by amorphous powders showed a larger dynamic strength and dynamic strain. The toughening mechanism with predominant transgranular fracture is explained based on the separation of quasi-boundaries.

## 1. Introduction

Al_2_O_3_ ceramics [1,2,3] are well known for their superior properties, including excellent high-temperature mechanical strength, resistance to chemical corrosion, good wear resistance and oxidation resistance. Moreover, compared with other ceramic materials, Al_2_O_3_ has the advantages of abundant resources, large reserves, and low prices [4,5]. Therefore, Al_2_O_3_ ceramics are the most widely employed oxide ceramics in industry, including aerospace, automotive, chemical and medical, metallurgy, nuclear technology, electrochemical devices, optoelectronic and armor protection [6,7,8,9,10,11,12]. For instance, in the field of armor protection, the high degree of hardness and high strength of Al_2_O_3_ ceramics at high strain rates are conducive to resisting the penetration of high-speed armor-piercing ammunition, and the low density is beneficial to the lightweight design of armor protection systems [13].

However, to achieve the full density of Al_2_O_3_ ceramics, sintering is commonly performed at temperatures above 1600 °C due to their high melting point, which leads to a serious growth of grains and the performance deterioration of ceramics. For decades, sintering at low temperatures [14,15,16,17] has been performed in varieties of ceramics, in attempts to obtain bulk ceramics with full density as well as maintaining nanoscale grain size, attributed to the unique mechanical and functional properties of ceramics with grain sizes within the nanometer to sub-micrometer range. The majority of this research focuses on the optimization of sintering strategies, such as two-step sintering [18,19,20] or the employment of field-assisted sintering techniques, including hot pressing [21,22] and spark plasma sintering [23,24,25]. Apart from that, low temperature sintering assisted by amorphous powders was achieved in nanoceramics by virtue of the transition from the amorphous to crystalline phase [26,27,28,29,30]. However, amorphous Al_2_O_3_ starting powders are seldom reported to our knowledge.

Al_2_O_3_ ceramics are also restricted by their brittleness, and several techniques have been commonly reported to enhance the toughness. To improve the toughness of Al_2_O_3_ ceramics, ZrO_2_ powders are commonly doped as the toughening phase attributed to the transformation toughening [31,32] from t-ZrO_2_ to m-ZrO_2_ with shear strain of 3~5% or volume expansion of 8%, which significantly hinders the crack propagation and improves the fracture energy. Another mechanism is nanoparticle strengthening [33,34], which employs nanoparticles as the second phase and disperses them in the ceramic matrix, utilizing the advantages of nanoparticles to improve the performance of the composite material and achieve high strengthening and toughening. Ceramics are also reinforced by fibers or whiskers [35,36,37,38], in which fibers or whiskers can prevent the propagation of cracks and enhance the fracture energy by crack deflection, whisker extraction and whisker bridging. Distinct to those mechanisms, a new toughening with a predominant transgranular fracture mode of nanograined ZrO_2_ ceramics was approved by Shen [39], and the transgranular fracture was attributed to the disassembling of mesocrystalline grains.

In this work, Al_2_O_3_ nanoceramics were fabricated by spark plasma sintering, and the powders selected as the starting powders were both amorphous and nanocrystalline. Compared to nanocrystalline powders, the amorphous powders were consolidated into full density Al_2_O_3_ nanoceramics at a low temperature, and the samples exhibited superior static and dynamic mechanical properties. Furthermore, the toughening mechanism with predominant transgranular fracture is explained based on the quasi-boundaries.

## 2. Materials and Methods

### 2.1. Powder Preparation

Amorphous Al_2_O_3_ powders were prepared by the sol–gel method. Aluminum nitrate (AR, Aladdin, Shanghai, China) and citric acid (AR, Aladdin, Shanghai, China) were dissolved in deionized water and stirred in a water bath at 70 °C, during which the solution became a colloid due to the evaporation and polymerization. The colloid was then put into a baking oven at 120 °C, and transformed into a porous gel after burning. Calcination of the porous gel was performed in a furnace at 600 °C for 1 h under atmosphere pressure, and amorphous Al_2_O_3_ powders were obtained. Furthermore, to investigate the effect of different Al_2_O_3_ powders on sintering, commercial Al_2_O_3_ powders (50 nm, >99.99% pure; Sumitomo, Tokyo, Japan) with a similar particle size were employed as the referential powder.

### 2.2. Spark Plasma Sintering

To consolidate the Al_2_O_3_ powders into bulk dense ceramics, spark plasma sintering was carried out using a DR. SINTER, SPS-2050 apparatus (Fuji Electronic Industrial, Kanagawa, Japan). The molds employed in SPS sintering were made of graphite, with an inner diameter of 20 mm. Considering that the powders easily adhered to the graphite molds during sintering, graphite papers with thickness of 0.3 mm were employed to wrap up the powders.

For each sample, the powders were added into the graphite mold and cold-pressed at 10 MPa, and then the graphite die was placed into the SPS apparatus. After loading of 75 MPa, the apparatus was heated to sintering temperature of 900~1500 °C with heating rate of 100 °C/min under axial compression of 50 MPa in vacuum atmosphere under pressure of below 4 Pa. The temperature and pressure were held for 5 min, and the Al_2_O_3_ ceramics were obtained after cooling and demolding.

### 2.3. Test and Characterization

Archimedes’ method based on ASTM C373 was conducted to determine the bulk density and porosity of the ceramics after SPS. X-ray diffraction analysis (XRD, D/Max-2500v/pc, Rigaku, Tokyo, Japan) with Cu Kα radiation was performed to characterize the phase composition of Al_2_O_3_ powders and ceramics after SPS sintering. Scanning electron microscope (SEM, Model S4800; Hitachi, Tokyo, Japan) and HRTEM (FEI Tecnai G2 F20, Hillsboro, OR, USA) were carried out to examine the morphology and particle size of the powders and sintered ceramics. To ensure the pictures were stable, the powders underwent sufficient ultrasonic dispersion and spray gold treatment before the SEM test. ^27^Al magic angle spinning nuclear magnetic resonance (^27^Al MAS-NMR) was carried out to investigate the short-range structures of the powders, and the spectra were collected by Infinityplus 300 spectrometer (Varian, Palo Alto, CA, USA) at 104 MHz with chemical shifts referenced to external.

Vickers indentation tests were carried out on the polished surfaces of sintered alumina samples using a microhardness machine (HXD-1000TM, Shanghai, China) to study the Vickers hardness (H_V_) of the samples, during which 10 N was loaded and held for 15 s. Based on the diagonal length of indentation, the H_V_ was evaluated through the equation [40]:(1)$$H_{V}=1.854\frac{P}{d^{2}}$$
where P presented the load, and d expressed the mean value of the diagonal length. Each value of H_V_ took an average value of 20 indentations.

Three-point bending was carried out to measure the flexure strength (σ) of the samples using an Instron5500R electron mechanical universal material testing machine (Boston, MA, USA). The sintered bars were in the dimension of 1.5 × 2 × 20 mm, the supporting span of 15 mm, with a loading rate of 1 mm/min. The flexure strength (σ) was determined based on the equation [41]:(2)$$\sigma =\frac{3\mathrm{PL}}{2bd^{2}}$$
where P was the load at which the bars broke down; L was the support span; b was the specimen width; and d was the specimen thickness. Each measurement was tested on 8 specimens, and the mean values of these measurements were taken as the three-point bending values.

The fracture toughness was also measured through single edge notched beam specimen techniques. The dimensions of the testing bars were 30 × 6 × 4 mm in length, thickness and width, with notch depth of 2.5 mm, and the supporting span in three-point bending was 20 mm. The fracture toughness through SENB method could be expressed by [42]:(3)$$K_{\mathrm{IC}}=Y\frac{3\mathrm{PL}}{2bh^{2}}\sqrt{a}.$$
(4)$$Y=1.99-2.47\frac{a}{h}+12.97{\left(\frac{a}{h}\right)}^{2}-23.17{\left(\frac{a}{h}\right)}^{3}+24.80{\left(\frac{a}{h}\right)}^{4}$$
where P was the largest load before fracture; L was the supporting span, b; h was the length, width and thickness for the testing bars; and a was the notch depth. Each value was determined by taking the average of 7 specimens.

To explore the application of Al_2_O_3_ ceramics in an armor protection system, a split Hopkinson pressure bar (SHPB) was carried out on samples in the scale of Φ13 mm × 6 mm to investigate the dynamic mechanical properties. Based on the stress wave propagated in the pressure bar, the function of stress, displacement (strain) of the bar versus time was analyzed, and the dynamic stress–strain curves at different strain rates of the ceramics were obtained.

## 3. Results and Discussion

### 3.1. Powder Characteristics

XRD patterns of the sol–gel prepared and commercially purchased powders are displayed in Figure 1a. The powders prepared by the sol–gel method show only broad dispersion peaks for the amorphous phase without any diffraction peaks for crystalline phases. Considering that the sol–gel powders were prepared with raw materials of aluminum nitrate and citric acid, Al^3+^ is the only metal ion in the powders. As the powders were calcinated at 600 °C for 1 h under atmosphere, the calcination ensures the complete decomposition of organic compound and bound water. Therefore, only Al^3+^ and O^2−^ remains after calcination, and the lower curve corresponds to amorphous Al_2_O_3_. In comparison, the commercially purchased powders are well crystallized, and characteristic peaks identified as α-Al_2_O_3_ (PDF #42-1468) are detected in the XRD pattern.

To further investigate the short range structure of the different Al_2_O_3_ powders, ^27^Al MAS-NMR spectra are recorded from the sol–gel prepared and commercially purchased powders and displayed in Figure 1b. In the standard spectrum of α-Al_2_O_3_, aluminum atoms exist in the form of six coordination (^[6]^Al) located near 12 ppm, and the aluminum atoms in γ-Al_2_O_3_ are in the form of ^[6]^Al near 8 ppm and ^[4]^Al near 66 ppm [43]. In the nuclear magnetic resonance spectrum in Figure 1b, only one resonance peak is detected at about 12 ppm for the commercially purchased powders, which is identified as ^[6]^Al. Therefore, the aluminum atoms in the commercially purchased powders are in the form of α-Al_2_O_3_, which is consistent with the XRD spectrum in Figure 1a. The aluminum elements in sol–gel prepared powders exhibit a mixture of ^[4]^Al (53 ppm), ^[5]^Al (29 ppm) and ^[6]^Al (5 ppm), and the specific positions of the ^[4]^Al and ^[6]^Al peaks are slightly different from the standard peaks in α-Al_2_O_3_ or γ-Al_2_O_3_. The mixture of ^[4]^Al, ^[5]^Al and ^[6]^Al is attributed to the almost random distribution of Al ions in the O ion vacancies in amorphous Al_2_O_3_, whose elements are arranged more freely and have a higher degree of disorder than the crystalline phases. The amorphous phase for the sol–gel prepared powders also agrees with the XRD spectrum in Figure 1a.

SEM micrographs taken on the sol–gel prepared and commercially purchased powders are displayed in Figure 2. The amorphous Al_2_O_3_ powders prepared by the sol–gel method in Figure 2a are near round in shape, and the agglomeration is slight with weak contacts between the particles. The mean particle size is about 40 nm, and the particle size is relatively uniform with a narrow span of distribution. In Figure 2b, the commercially purchased Al_2_O_3_ powders show not only spherical shapes but also nonspherical particles. The straight contours exhibited in the nonspherical particles suggest that the powders are in well crystallization. Compared to the sol–gel prepared powders, the commercially purchased powders show a similar degree of aggregation and an approximate average particle size of 40 nm.

### 3.2. Spark Plasma Sintering

During the SPS sintering of the different powders, the displacement of the mold plunger was recorded versus the temperature, and plotted in Figure 3. Apparently, the displacement of the mold plunger in SPS is related to the structural densification of ceramics as a result of the multiple mechanisms, including mass transfer, vacancy diffusion and particle rearrangement. In the SPS of sol–gel prepared amorphous Al_2_O_3_ powders, the displacement curve of the mold plunger is approximately an S-shape. The displacement is negligible below 1100 °C, and the displacement growth mainly takes place in the period of 1150~1350 °C from 0.5 mm to 6.5 mm, after which the displacement increases mildly to the maximum value of about 7.2 mm at 1500 °C. It is suggested that the mass transfer and vacancy diffusion are highly limited below 1100 °C, and become prominent at 1150~1350 °C. During the SPS of commercially purchased Al_2_O_3_ polycrystalline powders, the displacement curve is approximate to the J-curve, which is significantly different from that of amorphous powders. The value of displacement grows to 2.3 mm below 1350 °C, and a significant growth is continued during 1350~1500 °C resulting in a maximum displacement of 6.0 mm. The significant differences in the displacement curves are attributed to the better sintering activity of amorphous Al_2_O_3_ powders. For the amorphous powders, the densification mainly takes place in the process of 1150~1350 °C, and the densification is almost completed at 1350 °C. In comparison, for the commercial polycrystal powders, the densification occurs during 1350~1500 °C, and the densification is uncompleted even at 1500 °C. Therefore, the displacement curve for sol–gel prepared amorphous Al_2_O_3_ powders is approximate to an S-shape, but approximate to a J-curve for commercial polycrystalline powders.

To further understand the densification behavior of different powders during SPS, the derivative of displacement (dx/dT) versus the SPS temperature for different powders is performed and plotted in Figure 4. For the sol–gel prepared amorphous Al_2_O_3_ powders, the majority of the structural densification occurs between 1150 °C and 1350 °C, and the maximum value of dx/dT is 0.05 mm/°C at about 1270 °C. In contrast, during the SPS of commercially purchased polycrystalline powders, the structural densification mainly takes place between 1250 °C and 1500 °C, and a considerable densification is shown above 1400 °C. The maximum value of dx/dT is less than 0.035 mm/°C when adjacent to 1350 °C. It is further confirmed that the amorphous Al_2_O_3_ powders have better sintering activity than the commercial polycrystal powders, and they are easier to convert into dense volumes at low temperatures. Compared to polycrystal powders, amorphous powders are in a metastable state with high free energy. In this condition, the content of point defects is much higher in amorphous powders, which accelerates the diffusion of defects and particle rearrangement during the SPS sintering.

After the Al_2_O_3_ ceramics are SPS sintered at different temperatures for 5 min, the relative densities are measured and plotted versus the SPS temperature in Figure 5. The relative density of 98% is represented by the dash line, above which the samples are regarded as being near-full density. During the SPS sintering of commercial Al_2_O_3_ crystalline powders, the relative density of the bulk sample increases monotonically from 75.8% to 98.7% with increasing temperature. The relative density is not higher than 98% until the SPS temperature is up to 1450 °C. In comparison, the sample of the sol–gel prepared amorphous powders show a larger relative density than those with commercial powders at any temperature ranging from 1200 °C to 1450 °C. During the SPS sintering of amorphous Al_2_O_3_ powders, the relative density is 89.2% at 1200 °C, and the value increases to 98.3% as the SPS temperature is 1300 °C. To ensure a near-full density higher than 98%, the threshold temperature in SPS is 1450 °C for commercial Al_2_O_3_ crystalline powders, and decreases to 1300 °C for amorphous powders. Therefore, the amorphous powders highly improve the structural densification of Al_2_O_3_ ceramics at low temperatures, compared to the commercial polycrystalline powders, which is attributed to the better sintering activity of amorphous Al_2_O_3_ powders, in agreement with Figure 3 and Figure 4.

### 3.3. Microstructure

Al_2_O_3_ ceramics with a relative density above 98% are fabricated by SPS sintering of commercial polycrystalline powders at 1450 °C and sol–gel amorphous powders at 1300 °C. The SEM micrographs are taken on the natural surface of as-sintered Al_2_O_3_ ceramics and displayed in Figure 6. After the SPS sintering of commercial polycrystalline Al_2_O_3_ powders, the sample shows dense packing of particles with almost no pores in Figure 6a, which is related to the porosity of about 1.3%. The particle size shows a broad distribution, and the mean particle size is about 350 nm. After SPS sintering of amorphous Al_2_O_3_ powders, the ceramic shows a reasonably dense microstructure with almost no detectable pores in Figure 6a, which is in agreement with the low porosity of 1.7%. Moreover, the nanograined microstructure is very homogeneous with a mean size of about 170 nm.

SEM images are also taken on the fracture surface of SPSed Al_2_O_3_ ceramics and shown in Figure 7. The sample by SPS of commercial polycrystalline powders in Figure 7a shows an intergranular fracture feature, in which faceted grains morphology is revealed. Even though the microstructure on the fracture surface is still very dense, the grain size is slightly smaller than that on natural surface. It is reasonable that the temperature is slightly higher on the natural surface than the inside volume, which leads to easy atomic diffusion on the surface and promotes grain growth. Distinct from the intergranular fracture morphology, the sample SPSed from sol–gel amorphous powders shows a mixed morphology with both intergranular and transgranular fracture in Figure 7b, where the transgranular fracture areas are marked by red circles. The rough fracture surfaces with streaks across the individual grains are detected and quantities of microcracks are shown among the Al_2_O_3_ grains. Generally speaking, intergranular facture is the common fracture mechanism in nanograined ceramics, due to the high strength of nanoparticles [44,45]. The transgranular fractures in nanograined ceramics are seldom reported on, and this fracture mode can be responsible for special mechanical properties.

### 3.4. Mechanical Properties

The mechanical properties along with relative densities of the Al_2_O_3_ ceramics prepared with different powders are listed in Table 1. It is shown that the Al_2_O_3_ ceramics prepared with sol–gel amorphous powders display superior mechanical properties, including a high bending strength of 870 MPa, Vickers hardness of 20.5 GPa and fracture toughness of 4.3 MPa∙m^1/2^. Even though the Al_2_O_3_ ceramics prepared with sol–gel amorphous powders have a slightly lower relative density and hardness, their bending strength and fracture toughness are much higher than those with polycrystalline powders. It is revealed that the amorphous powders prepared by sol–gel method as starting powders contribute to the high bending strength and fracture toughness, and have weak impact on the hardness.

Dynamic mechanical behaviors of the Al_2_O_3_ ceramics are measured using SHPB, with strain rates of 300 s^−1^, 600 s^−1^ and 900 s^−1^, respectively, and the stress–strain curves are obtained and shown in Figure 8a,b. The strain rate versus time is also plotted in Figure 8c to show the stable strain rate, during the period of about 47~49 μs the strain rate approximates to a constant value of 900 s^−1^, indicating that the specimen deforms at a constant strain rate. The major deformation mode of Al_2_O_3_ ceramics is elastic deformation, and the nonlinear deformation is caused by the main crack during impact loading, rather than plastic deformation. With the increase in strain rate, the dynamic compressive strength and strain show a monotonic increase. Crack propagation is the dominating mode for the Al_2_O_3_ ceramic, which is known as a brittle material. As the crack propagation in ceramics is slower than the external loading rate, the cracks are not able to propagate in time and delay the failure of ceramics as a consequence. Therefore, the Al_2_O_3_ ceramics display higher compressive strength and deformation strain at higher strain rates. For each strain rate, the ceramics prepared with amorphous powders display higher dynamic strength and strain than those with commercial polycrystalline powders. At the strain rate of 900 s^−1^, the dynamic strength and strain are 3850 MPa and 3.9% for ceramics prepared with amorphous powders, and the values are 3280 MPa and 3.3% for ceramics prepared with commercial polycrystalline powders.

The dynamic strength can be attributed to the hindrance of grain boundaries to the dislocation movements. As dislocations move to grain boundaries, greater force is required to overcome their hindrance, thereby allowing the deformation of the grain to transfer to another grain. Compared to the Al_2_O_3_ ceramics with commercial polycrystalline powders, the ceramics prepared with amorphous powders are a smaller grain size, and have a higher proportion of grain boundaries. Therefore, the greater the hindrance effect on dislocation movement, the higher its strength is, which is consistent with the strength in static tests. The dynamic strain can be attributed to the hindrance effect of toughness on cracks. The dynamic deformation is mainly composed of elastic deformation, the formation and propagation of cracks and the continuous accumulation of damage. The dynamic strain related to the highest point of the stress−strain curve is actually the elastic deformation strain before the formation of the main crack. It is suggested that the Al_2_O_3_ ceramics prepared with amorphous powders have a greater toughness, which hinders the formation and propagation of main cracks, delays material failure, and exhibits larger dynamic strain.

### 3.5. Toughening Mechanism

It is meaningful that the Al_2_O_3_ ceramics prepared with amorphous powders show superior mechanical properties, including bending strength, fracture toughness, dynamic strength and dynamic strain, compared to samples with commercial polycrystalline powders. In this work, the superior mechanical properties are highly dependent on the unusual fractography with transgranular fractures in nanograined ceramics.

To explore the transgranular fractography, Al_2_O_3_ ceramic SPSed with amorphous powders are polished by Ar ion beam, and the SEM backscattered electron image is shown in Figure 9a. It is obvious that, apart from the contrasts between different grains, there are obvious contrasts within the individual particles. It is known that the contrasts in SEM backscattered electron images are related to the variation in crystallographic orientation or chemical composition. Considering that the raw powders are Al_2_O_3_ without any impurity elements, the contrasts within the individual particles probably resulted from the difference in crystallographic orientation.

To further understand the crystallographic orientation in the Al_2_O_3_ particles, a TEM image was taken within an individual grain and is displayed in Figure 9b. According to the lattice fringes, the individual grain is composed of smaller nanocrystallites with different crystallographic orientations, even though individual grains in nanoceramics is regarded as a single crystal with uniform crystallographic orientation in traditional cognition. Between the nanocrystallites with different crystallographic orientations, there are some boundaries with large angles (marked by rectangles in Figure 9b), similar to those reported in normal ceramics. Furthermore, another kind of boundary with small angles is detected and marked by ellipses, which is defined as a quasi-boundary.

In traditional cognition, individual grains in nanoceramics are regarded as single crystals with uniform crystallographic orientation, and it is quite difficult to break the single crystals through propagation of cracks. That is why intergranular fracture is the normal fracture mode in nanoceramics, and transgranular fracture is seldom reported. It is suggested that the quasi-boundaries result from the mismatch of the lattice between two adjacent nanocrystallites within an individual grain, and they are probably responsible for the transgranular fractography in Figure 7b. In this work, the Al_2_O_3_ ceramic with superior mechanical properties is toughened by the abundant quasi-boundaries within individual grains. In the process of fracture, the mismatch of the lattice between two adjacent nanocrystallites becomes larger with the growth of cracks, leading to the breaking of quasi-boundaries. In this condition, the nanocrystallites adjacent to the quasi-boundaries separate with each other, and result in transgranular fracture modes. The broken quasi-boundaries serve as weak interfaces for crack propagation and deflection, consuming a larger quantity of fracture energy in failure, and resulting in enhanced strength and toughness.

## 4. Conclusions

In this work, Al_2_O_3_ amorphous powders and polycrystalline powders with similar particle sizes were employed as starting powders in preparation of Al_2_O_3_ nanoceramics by spark plasma sintering. The effective comparison of sintering processes and ultimate products depending on starting powder conditions was explored. To ensure near full density higher than 98% of the Al_2_O_3_ nanoceramics, the threshold temperature in SPS is 1450 °C for polycrystalline Al_2_O_3_ powders and 1300 °C for amorphous powders, and the mean grain sizes are 350 nm and 170 nm, respectively. The easier densification at a low SPS temperature for amorphous powders is attributed to the metastable state with high free energy of amorphous powders. The Al_2_O_3_ nanoceramics prepared by amorphous powders display superior mechanical properties, including a high bending strength of 870 MPa, Vickers hardness of 20.5 GPa and fracture toughness of 4.3 MPa∙m^1/2^. Furthermore, the Al_2_O_3_ nanoceramics prepared by amorphous powders showed larger dynamic strength and dynamic strain. The toughening mechanism with predominant transgranular fracture is explained based on the separation of quasi-boundaries.

## Figures and Tables

**Figure 1 materials-16-05652-f001:**
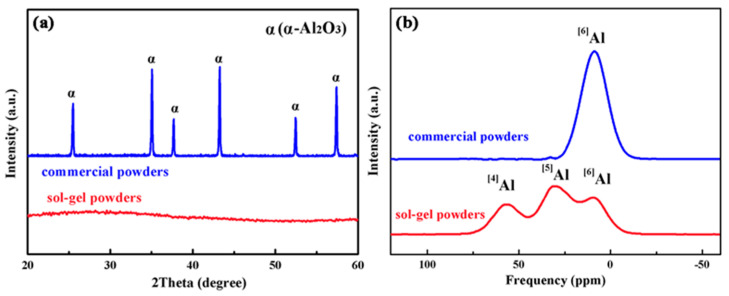
(**a**) XRD patterns and (**b**) ^27^Al MAS-NMR spectra recorded from the sol–gel prepared and commercially purchased powders.

**Figure 2 materials-16-05652-f002:**
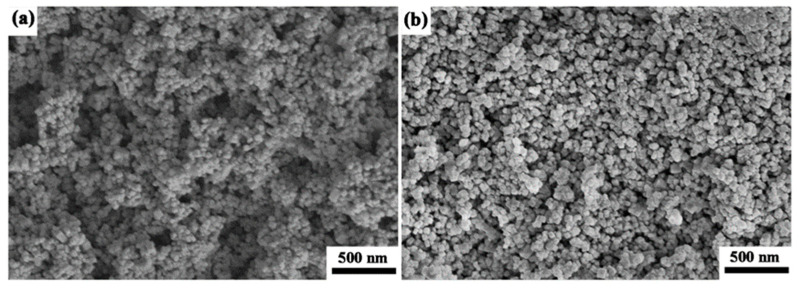
SEM micrographs of the (**a**) sol–gel prepared and (**b**) commercially purchased powders.

**Figure 3 materials-16-05652-f003:**
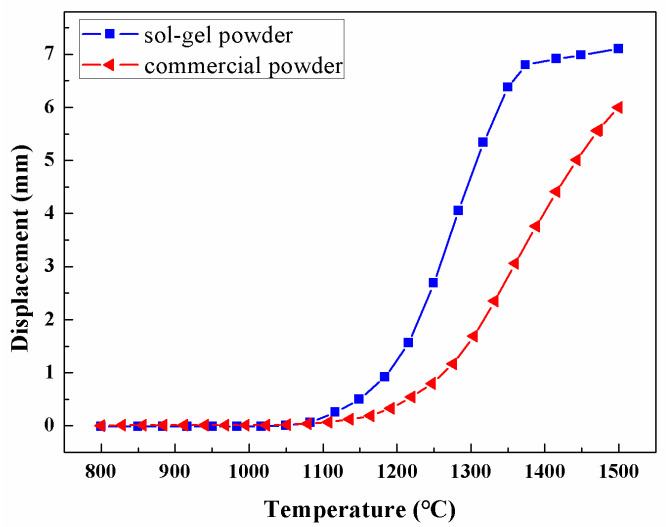
Displacement of the mold plunger with the increasing temperature in SPS of different Al_2_O_3_ powders.

**Figure 4 materials-16-05652-f004:**
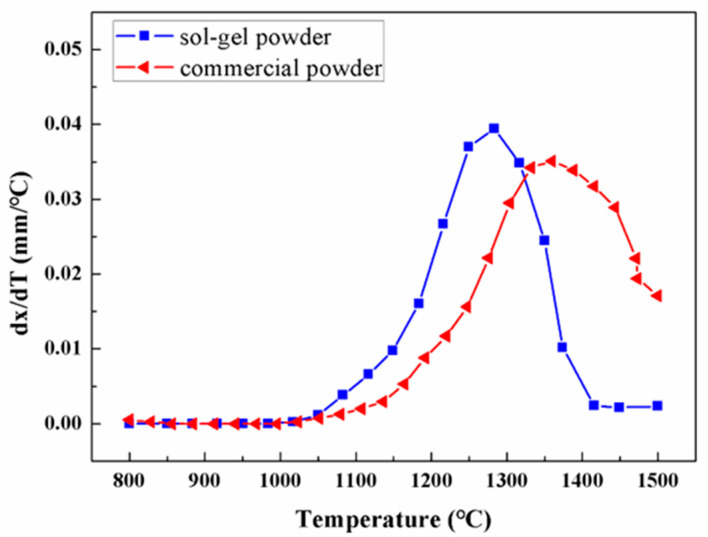
The derivative of displacement versus SPS temperature for different Al_2_O_3_ powders.

**Figure 5 materials-16-05652-f005:**
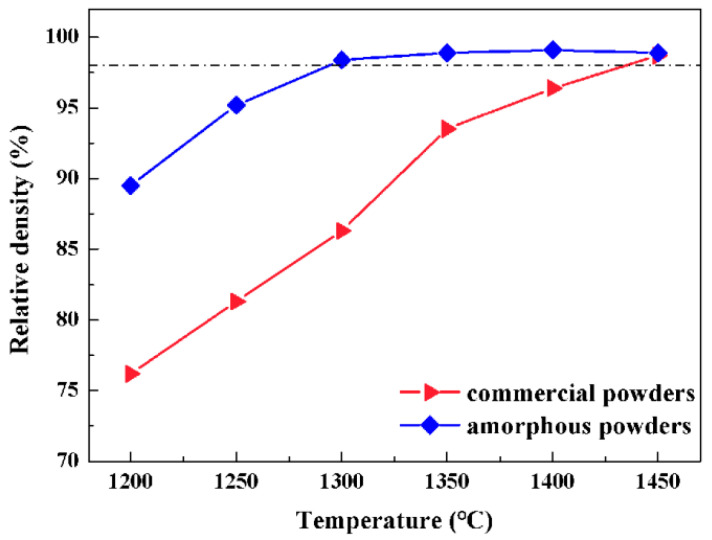
The relative density of Al_2_O_3_ ceramics by SPS at different temperatures.

**Figure 6 materials-16-05652-f006:**
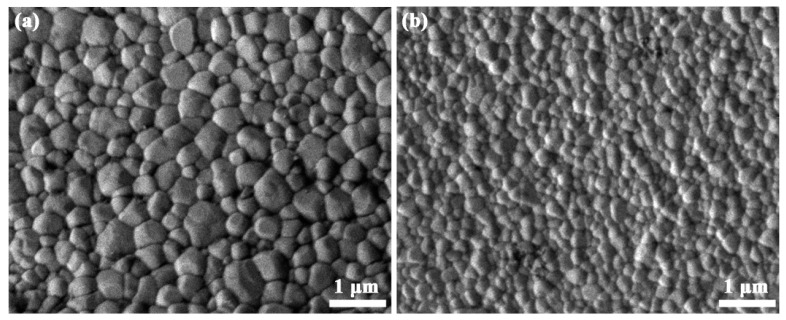
SEM images of the natural surfaces from Al_2_O_3_ ceramics by SPS of (**a**) commercial polycrystalline powders and (**b**) sol–gel amorphous powders.

**Figure 7 materials-16-05652-f007:**
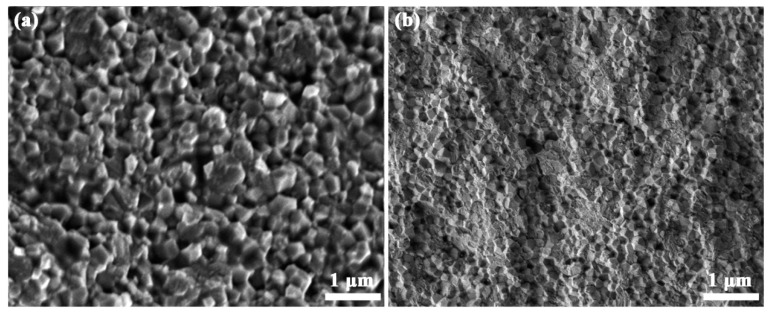
SEM images of the fracture surfaces from Al_2_O_3_ ceramics by SPS of (**a**) commercial polycrystalline powders and (**b**) sol–gel amorphous powders.

**Figure 8 materials-16-05652-f008:**
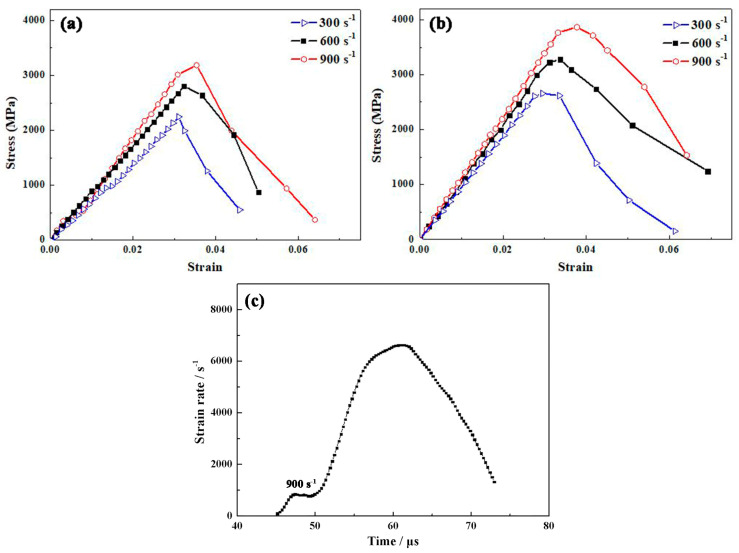
Dynamic stress−strain curves under different strain rates for Al_2_O_3_ ceramics by SPS of (**a**) commercial polycrystalline powders and (**b**) sol−gel amorphous powders; (**c**) strain rate plotted versus time to show the stable strain rate.

**Figure 9 materials-16-05652-f009:**
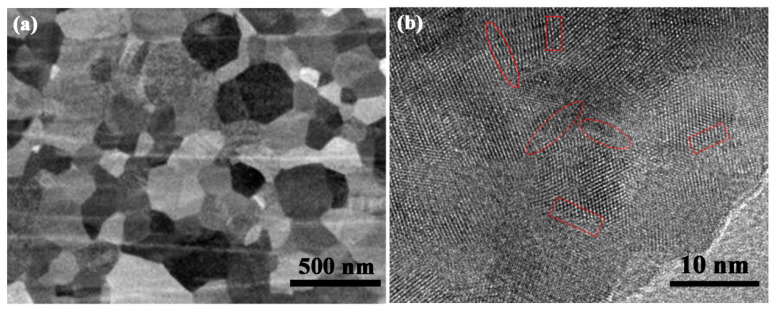
(**a**) SEM images after polishing using Ar ion beam and (**b**) HRTEM images of the Al_2_O_3_ ceramic SPSed with amorphous powders.

**Table 1 materials-16-05652-t001:** Relative densities and mechanical properties of the Al_2_O_3_ ceramics prepared with different powders.

Raw Powders	Relative Density	Bending Strength (MPa)	Hardness (GPa)	Fracture Toughness (MPa∙m^1/2^)
sol–gel amorphous	98.3%	870 ± 55	20.5 ± 1.2	4.3 ± 0.35
Commercial	98.7%	710 ± 78	21.3 ± 1.7	2.9 ± 0.32

## Data Availability

Not applicable.

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
