# Peer review of "Enhanced Mechanical Properties of Al2O3 Nanoceramics via Low Temperature Spark Plasma Sintering of Amorphous Powders"

_materials, 2023, doi:10.3390/ma16165652_

Round 1
Reviewer 1 Report
The following comments need to be addressed for the paper entitled "Enhanced mechanical properties of Al2O3 nanoceramics via low-temperature spark plasma sintering of amorphous powders", before publication.
1) During the sintering process, why do authors use graphite papers with a thickness of 0.3 mm?
2) Generally, the split Hopkinson pressure bar (SHPB) is used only to calculate a high strain rate. What are the applications of Al2O3 ceramics for SHPB?
3) Line 166 - J-curve, which is significantly different from that of amorphous powders. What is the reason?
4) The authors mentioned SHPB in the manuscript. But not discussed in detail.
5) Add the SHPB sample size and strain rate curve values.
Dear Editor,
The manuscript will accept after the modification
Thanks & Regards
Author Response
Dear Editor and Reviewers,
We have tried our best to revise and improve the manuscript and made great changes in the manuscript according to your good comments. This file mainly explains how we have addressed the reviewer's concerns and the revised manuscript are marked at places where the content has been revised. We appreciate for your work earnestly, and hope that the corrections will meet with your approval.
Yours Sincerely
Xiqing Xu
-------------------------------------------------------------------------------------
The following is a point-to-point response to the comments.

Reviewer 2 Report
The manuscript is devoted to the study of the influence of amorphous powders on sintering processes and properties of Al2O3 ceramics. The topic is very relevant, allowing you to expand your understanding of SPS processes.
The topic is very relevant, allowing you to expand your understanding of SPS processes:
1. In the introduction, one of the paragraphs of the article is devoted to phase transformation toughening of ZrO2. The mechanisms considered are characteristic of zirconium dioxide, and not of aluminum oxide. In my opinion, the authors need to explain why they cite such a paragraph in the key of their research.
2. According to the text of the article, it is unclear whether we are talking about a mold with an inner diameter of 20 mm or an outer one. Please clarify, this is an important point.
3. The process of making samples and preparing the mold is not described in sufficient detail. The number of samples per one point of the experiment is not specified.
4. Section 3 presents the results of NMR spectroscopy, but section 2 does not mention this method.
5. There are multiple inaccuracies and typos in the work. For example, line 70 repeats "in" twice in a row. Line 93 mentions the polished surface of ZrO2, although the manuscript is dedicated to Al2O3.
Author Response
Dear Editor and Reviewers,
We have tried our best to revise and improve the manuscript and made great changes in the manuscript according to your good comments. This file mainly explains how we have addressed the reviewer's concerns and the revised manuscript are marked at places where the content has been revised. We appreciate for your work earnestly, and hope that the corrections will meet with your approval.
Yours Sincerely
Xiqing Xu
-------------------------------------------------------------------------------------
The attachment is a point-to-point response to the comments.

Reviewer 3 Report
Referee report on “Enhanced mechanical properties of Al2O3 nanoceramics via low-temperature spark plasma sintering of amorphous powders”
This is a quite interesting and good paper that certainly can be recommended for publication, but clarifying and detailing some parts of the text.
1. Line 27. In the introduction, in order to increase the visibility of the article for a wide range of readers, it is important to note also other applications of ceramics on the base on Al2O3 , for example, in nuclear technology as a shielding materials, electrochemical devices or in optoelectronic:
A) Qi S., Porotnikova N.M., et al. Transactions of Nonferrous Metals Society of China, 2016, 26 (11), pp. 2916-2924. https://doi.org/10.1016/S1003-6326(16)64421-7
B) Popov, A. I., et al. Nuclear Instruments and Methods in Physics Research Section B: Beam Interactions with Materials and Atoms, 2018, 433, 93-97.
C) Ehrhart, P., et al. Radiation Effects and Defects in Solids, 1995, 136(1-4), pp. 169–173.
2. Fig. 1(a). Give a more detailed explanation why the lower curve (sol-gel) corresponds to Al2O3 and not something else.
3. Fig.2. How stable are these pictures and were there any indications for aging and how was it checked?
4. Fig. 3 and 4. Is this temperature range (1100 -1300 C) somehow related to the diffusion of vacancies?
5. How different are ceramics and powders in terms of the content of point defects and how can this affect the results?
6. How porous are the resulting ceramics?
7. In the conclusions, it is necessary to clearly formulate what new data about the studied materials were obtained in this work?
In general, the manuscript is interesting and can be considered for publication after constructive reflection on the above comments.
Author Response

(The authors gave the same response as above.)

Round 2
Reviewer 1 Report
The manuscript is accepted in its present form.
The manuscript is accepted in its present form.
Reviewer 3 Report
The authors have successfully improved their original manuscript, which now can be recommended for publication.